# The Use of Intrinsic Disorder and Phosphorylation by Oncogenic Viral Proteins to Dysregulate the Host Cell Cycle Through Interaction with pRb

**DOI:** 10.3390/v17060835

**Published:** 2025-06-10

**Authors:** Heidi Kast-Woelbern, Sarah K. Martinho, Kayla T. Julio, Audrey M. Vazzana, Abbey E. Mandagie, Ariane L. Jansma

**Affiliations:** 1Department of Biology, Point Loma Nazarene University, 3900 Lomaland Drive, San Diego, CA 92106, USA; heidiwoelbern@pointloma.edu (H.K.-W.); sarah.martinho6@gmail.com (S.K.M.); kaylatjulio@gmail.com (K.T.J.); 2Department of Chemistry, Point Loma Nazarene University, 3900 Lomaland Drive, San Diego, CA 92106, USA; audrey.m.vazzana@gmail.com (A.M.V.); abbeymandagie@gmail.com (A.E.M.)

**Keywords:** oncogenic virus, intrinsically disordered protein, phosphorylation

## Abstract

Approximately 15% of cancers worldwide are caused by oncogenic viruses. These infectious agents utilize multiple strategies to dysregulate their host cells as a means of viral reproduction. While this typically involves a small number of viral oncoproteins known to interact with a myriad of host cell proteins, direct binding with the tumor suppressor retinoblastoma protein (pRb) as a means to dysregulate the cell cycle appears to be a common mechanism among most known oncogenic viruses. This review evaluates the shared structural themes of binding motif, intrinsic disorder, and viral oncoprotein phosphorylation, utilized by eight different oncogenic viruses for the subjugation of pRb. Cancer caused by oncogenic viruses represents one of the few potentially preventable forms of cancer. The more we understand the common strategies used by these infectious agents, the better equipped we will be to further optimize vaccination and therapeutic strategies to fight them.

## 1. Introduction

In 1911, Peyton Rous published his work on a “filterable agent” (known to be smaller than bacteria) from the tumor extract of a hen that was transmissible to other fowl and capable of inducing tumorigenesis [1]. This foundational study provided evidence of an infectious agent, later determined to be a virus, causing cancer. Subsequently, innumerable labs have investigated the role of infectious agents, including viruses, in inducing cancer. Despite the vast array of human viruses, only a small subset have been definitively determined to be the causative agent for particular types of cancer (Table 1). However, approximately 15% of human cancer cases are currently attributable to this relatively small subset of viruses [2]. The International Agency for Research on Cancer (IARC) maintains a list of oncogenic infectious agents (viruses, parasites, and bacteria) defined as Group 1, which have been classified as carcinogenic to humans. This classification currently includes the following seven viruses: human immunodeficiency virus (HIV) type 1 (infection with), high-risk human papillomavirus (HPV), human T-cell lymphotropic virus type 1 (HTLV-1), hepatitis C virus (HCV), the Epstein–Barr virus (EBV), Kaposi sarcoma-associated herpesvirus (KSHV), also referred to as human herpesvirus type 8 (HHV-8) and hepatitis B virus (HBV). We have also included Merkel cell polyomavirus (MCPyV) and BK polyomavirus (BKPyV) to the list of oncogenic viruses due to their known association with cancer in immunosuppressed individuals. They are classified as Groups 2A and 2B, respectively, probable carcinogens, on the IARC list [3,4,5]. While the IARC does a commendable job of staying up to date on oncogenic pathogens, it is not necessarily exhaustive. For example, human T-cell lymphotropic virus type 2 (HTLV-2) has been linked to rare T-cell cancers but is not listed as Group 1 or Group 2 [6]. We therefore chose to focus this review on HTLV-1. Additionally, while adenovirus was the first virus shown to cause cancer in non-human animal models, and it has been used as a system to study oncogenicity in vitro, it is not a known causative agent for human cancers, so it is not included in this review [7]. Finally, we have not included HIV, even though it is often associated with various cancers and listed as Group 1 on the IARC list. Current research suggests that HIV itself is not the cause of cancer, but, rather, the virus induces immune suppression, which often results in a cancer diagnosis. In many cases, HIV infection allows other oncogenic viruses, such as HPV, EBV, or KSHV, to induce cervical cancer, Burkitt lymphoma, or Kaposi sarcoma, respectively, for those with advanced AIDS [8].

One hallmark for most of these oncogenic viruses is that they have a high seroprevalence, but only rarely induce cancer. These common viruses typically establish persistent infections, though EBV stands out by potentially causing cancer shortly after initial infection [9]. Most oncogenic viruses are DNA viruses (including HPV, MCPyV, BKPyV, EBV, KSHV, and HBV), with HCV, a single-stranded RNA virus, being a notable exception. HTLV-1, while an RNA virus, undergoes reverse transcription into DNA. These viruses typically rely on the host’s enzymes for DNA replication (DNA polymerase), as well as the host’s cellular machinery for protein synthesis (RNA polymerase and ribosomes), and assembly into new viral particles. If the virus can avoid immune surveillance, then it can potentially co-exist for decades with its host, setting up a dynamic interaction often resulting in the modification of normal cellular pathways by the viral proteins.

Oncogenicity ultimately comes as a result of modifying the host’s proteins that control DNA repair mechanisms, modification of proliferative activity by activation of oncogenes, inactivation of tumor suppressors, or inhibition of apoptosis [2,10]. By the time tumor formation begins, portions of the virus have often integrated into the host genome, and the virus is no longer considered transmissible. For example, in patients with an active low-risk (non-cancer causing) HPV infection, the DNA is maintained within the host as an episome. Due to immune surveillance, the virus is often eliminated over time. However, in some cases, latent HPV infections have been found lasting over 30 years [11]. Samples from patients with cervical cancer have shown portions of high-risk HPV, such as HPV16 and HPV18, integrated within the host genome [12]. Such integration is thought to avoid immune detection by the host, since the virus is no longer produced as virions. Additionally, HPV integration into the host chromosome has been shown to decrease expression of E2, a viral regulatory protein, resulting in increased expression of oncogenic proteins E6 and E7, which, in turn, dysregulate the host cell cycle, cause genetic instability, and drive cancer progression [13,14].

For many of these oncogenic viruses, tumorigenesis is partly due to dysregulation of the cell cycle. Over the years, a myriad of cellular targets have been identified as binding partners for oncogenic proteins expressed by these viruses. Major players in cell cycle control are often the targets of these oncogenic proteins, such as transcription coactivators like CREB-binding protein (CBP) and its homolog p300 [15,16], TGF-BETA signaling systems [17], or through the deactivation of tumor suppressors like p53 [18]. One specific tumor suppressor impacted by many oncogenic viruses is the retinoblastoma protein (pRb), and, often, its homologs p107 and p130 [19,20,21,22]. The process by which pRb controls the cell cycle has been well characterized over the years [23,24,25,26]. Hypophosphorylated pRb acts as a regulator of the G1 to S phase transition of the cell cycle by binding to the E2F family of transcription factors, which are responsible for regulating genes involved in cell cycle progression, DNA replication, and cell proliferation. Binding to pRb sequesters E2F and recruits histone deacetylases (HDACs) to E2F-responsive promoters, making DNA inaccessible for transcription, and thus blocking the activation of E2F target genes [27].

The structure of pRb consists of the N-terminal domain, two cyclin fold subdomains, A and B, which interact extensively to form the A/B pocket domain, and the C-terminal domain. The A/B pocket provides the primary binding site for the transactivation domain (TAD) of E2F, with the intrinsically disordered C-terminal domain of pRb playing an additional role in binding to the “marked box” (MB) domain of E2F [28,29,30,31]. During normal cell cycle progression, increasing concentrations of Cyclin D and cyclin-dependent kinases (CDK) 4/6, as well as Cyclin E and CDK2, lead to the progressive phosphorylation of pRb, which has 16 putative CDK phosphoacceptor residues [32]. Cyclin D utilizes what is known as an LxCxE motif to facilitate direct binding to pRb [33,34,35]. The LxCxE motif refers to a conserved sequence found within many known binding partners to pRb, where “x” represents any amino acid. In the case of Cyclin D, this motif binds to the pocket B region of pRb, resulting in phosphorylation from CDK. This triggers a conformational change to pRb, which ultimately results in the displacement of E2F, and subsequent transcription of genes necessary for DNA synthesis, ultimately causing the cell cycle to transition from G1 to S phase [36,37]. The resulting hyperphosphorylated pRb is inactive until a later time when it is reactivated by phosphatases and resumes its job of sequestering E2F [36,37,38,39,40,41]. This overall process is illustrated in Figure 1A.

Through the investigation of the oncogenic proteins expressed by these viruses, several patterns emerge in their methods for control of pRb activity and ultimate dysregulation of cell cycle progression. The hijacking strategies employed by these virally expressed proteins can involve displacement of E2F by direct binding to pRb, enabling the initiation of S phase independently of cellular signals and growth factors, as well as targeted pRb degradation (Figure 1B) [10,23,26,27,28,29,30,31,42]. Alternatively, viral oncoproteins have been shown to bind pRb and directly target it for degradation, possibly independently of E2F function (Figure 1C) [9,43,44,45]. Still other viral proteins can promote increased levels of the Cyclin/CDK complexes, causing increased phosphorylation, and subsequent inactivation, of pRb (Figure 1D) [46,47,48,49]. Studies of viral oncoproteins also indicate that these strategies can be used in combination for cell cycle control. This review focuses primarily on methods employed by viral oncoproteins involving direct interaction with pRb.

As summarized in Table 1, direct pRb binding can be accomplished through multiple strategies. The first is LxCxE-dependent, where the virus utilizes an oncoprotein with an LxCxE motif like that of Cyclin D, capable of similar interaction with the pocket B domain of pRb and thus similar E2F displacement, but also often resulting in pRb degradation. This is the case for high-risk HPV16 E7 (as well as E7 from most high-risk HPV genotypes), BKPyV LTAg, and MCPyV LTAg (Table 1) [42,50,51,52,53]. Alternatively, degradation of pRb can be accomplished through interaction with an LxCxE-mimic motif, where a sequence with a similar pattern is found to bind to pRb and target it for degradation, evolved specifically by HTLV-1 Tax, HCV NS5B, and EBV EBNA-3C [43,44,45,54]. At the same time, direct interaction with pRb can also occur independently of a known LxCxE motif, as seen with KSHV LANA, which binds pRb in a region rich in leucine and glutamic acid, but no traditional hallmarks have yet been identified for LxCxE-dependent interaction [49]. Additionally, while HBx from HBV has not been shown to dysregulate the cell cycle through direct interaction with pRb, this oncoprotein shares distinct structural characteristics with the other oncoproteins in this review, providing a potential basis for future studies. Finally, it is worth noting that some viral oncoproteins, such as HTLV-1 Tax and EBV EBNA-3C, are known to be involved in more than one of these pathways, reinforcing the idea that viral oncoproteins often employ multiple strategies in the dysregulation of the cell cycle. Closer examination of the more characterized systems, such as high-risk HPV E7, suggests that optimal interaction with pRb appears to involve a degree of structural flexibility, often associated with intrinsic disorder, on the part of the vial oncoprotein [55,56].

Intrinsically disordered proteins (IDPs) and regions (IDRs) within proteins provide high flexibility and versatility. The use of intrinsic disorder has often been seen in association with viral proteins, giving them the structural flexibility to bind to an array of partners within their cellular hosts, while at the same time, utilizing minimal genetic material [57,58,59]. Since IDPs and IDRs lack formal secondary structure, their amino acid sequence, which dictates their state of disorder, is even more crucial to their functional capabilities compared to that of a folded protein [60]. As such, post-translational modifications, such as phosphorylation, have the propensity to significantly change their energetic landscape and, in some cases, drive their interactions with other proteins [61,62]. Table 1 shows that the proteins with LxCxE and LxCxE-mimic motifs tend to have phosphoacceptor amino acids near that region, and phosphorylation at those sites within the oncoprotein is often shown to enhance its ability to bind pRb and dysregulate the cell cycle [35,63,64,65].

Viral protein subjugation of host cell kinases to promote viral function has been observed for many years [66]. While studies of the specific host cell kinases responsible for phosphorylating these oncoproteins are ongoing, Casein Kinase 2 (CK2) has been identified or predicted for most of the oncoproteins containing an LxCxE motif, and some with a possible LxCxE-mimic motif (Table 1) [63,65,67,68,69,70]. CK2 is ubiquitously expressed, constitutively active, and responsible for the phosphorylation of many physiological substrates [71,72]. As such, it is known to be targeted by viral proteins to enhance their functionality [73]. In order to phosphorylate its substrates, CK2 requires acidic residues (D/E) at the +1 and +3 positions relative to the phosphoacceptor serine(s) [71,74]. Interestingly, LTAg from MCPyV is one of the few LTAg proteins that does not have this canonical sequence near the pRb-binding motif, and while some preliminary data and NetPhos3.1 analysis suggest MAPK as a potential kinase, it is not definitively known at this time [70]. Additionally, Akt kinase (also known as protein kinase B) has been identified as the kinase responsible for the phosphorylation of HCV NS5B in the vicinity of the LxCxE-mimic domain. While this work is ongoing, it has been shown that phosphorylation at this residue is necessary for optimal protein function, while its connection to pRb binding is currently unknown [75]. Akt kinase is not constitutively active, but it is often a target of viral proteins and thus represents a potential area for future research [76].

While many studies have been performed looking at the structural motifs surrounding regions of viral oncoproteins known to interact with pRb, we sought to combine these structural analyses with Predictor of Natural Disordered Regions (PONDR) scores for the purpose of highlighting previously reported structural patterns surrounding pRb-binding domains. Additionally, we are labeling phosphoacceptor serine residues near the pRb-binding region that have been previously shown to be important for pRb binding and/or oncogenic activity, noting, when available, the host cell kinases they subjugate in this process. As structure/function patterns continue to emerge, and viruses continue to evolve, it is crucial to monitor the specific mechanisms of common host cell interactions, such as pRb, among these oncogenic viral proteins. While we recognize that this is a small piece of a very large puzzle, the overall purpose of this review is to assist in the observation of potential patterns involving intrinsic disorder and phosphorylation in terms of optimal pRb interaction, with the hope of guiding new potential research directions.

## 2. LxCxE-Mediated pRb Interaction: Human Papillomavirus, Merkel Cell Polyomavirus, and BK Polyomavirus

### 2.1. Human Papillomavirus (HPV) E7 Directly Binds and Deactivates pRb Through LxCxE Motif

Human papillomavirus (HPV) is part of a family of small non-enveloped DNA viruses possessing icosahedral capsids that measure 52–55 nm in diameter. Their genetic structure consists of circular double-stranded DNA (dsDNA) about 8 kb in size. Within the genome, there are typically eight protein-coding sequences, also known as open reading frames (ORF) [77]. Among the papillomavirus family is HPV, which presently has more than 200 known genotypes [78]. HPV is a common sexually transmitted infection that can be classified as either high-risk or low-risk based on the capacity for inducing cancer. Low-risk HPV genotypes, the most common of which is HPV6b, are associated with benign skin lesions, such as warts [79]. High-risk genotypes of HPV can cause a number of different cancers, with cervical cancer being the most common, but also including penile, oropharyngeal, anogenital, and a recent potential connection to bladder cancer [80,81,82]. HPV is responsible for 4.5% of global cancer cases, most notably cervical cancers, for which high-risk HPV is associated with 99.7% of infections [80,83,84]. The high-risk HPV16 and HPV18 genotypes are responsible for 73% of these cancer cases [80]. The most effective preventative measure against HPV infection is vaccination [85]. However, these HPV vaccines are not currently universally available and, as a result, developing countries with limited access to vaccinations experience more than 20% of HPV-attributed cancers, whereas developed regions such as Australia, New Zealand, and the United States have less than 3% of cancers tied to HPV [80,83,84].

The genome for high-risk HPV genotypes encodes for two early oncogenic proteins, E6 and E7, that dysregulate the normal host cell cycle [86,87]. Specifically, the oncoprotein E7 binds pRb and displaces E2F while also targeting pRb for degradation, thus initiating the cell cycle independently of normal growth signals (Figure 1B) [50,86,88]. The ability of E7 to both displace E2F and target pRb for degradation has been studied extensively over the years [88,89,90,91,92]. At the same time, E6 works synergistically to hijack the regulation of p53 and thus inhibit apoptosis [93,94,95]. This combined effort results in uncontrolled cell growth, ultimately leading to cancer [96].

The protein E7 is relatively small, with the most prevalent high-risk HPV16 E7 containing 98 amino acids, and an average size of approximately 100 amino acids across all genotypes. The protein has three conserved regions, CR1, CR2 and CR3. The C-terminal CR3 domain has a zinc finger structure and provides the interface for homodimerization [97,98,99,100]. At the same time, the N-terminal half of the protein, consisting of CR1 and CR2, is intrinsically disordered, classifying E7 as an IDP [56,99,101,102,103]. The disordered CR2 region of the protein includes the LxCxE motif, which is the primary site for binding to pRb [55,98,104]. Similar to the Cyclin D/CDK complex, the LxCxE motif of E7 binds in an extended conformation to the B subdomain in the A/B pocket region of pRb [105]. While it has been shown that the LxCxE motif of E7 is necessary for high affinity binding to pRb, comparing it to the structure for the pRb A/B pocket bound to the TAD region of E2F, the binding domains within pRb do not overlap, suggesting that they are able to bind independently (Figure 2A, alignment of structures adapted from PDB: 1n4m and PDB: 1gux, respectively) [31,105]. Therefore, the LxCxE motif is not sufficient on its own to displace E2F from pRb in cells [89]. At the same time, the structured CR3 domain of E7 was demonstrated to bind the C-terminal domain of pRb in a region overlapping with the lower affinity E2F-binding site [31,98]. Further studies revealed that the CR3 domain of E7 is also able to bind the MB domain of E2F [31]. E7 control of the cell cycle therefore requires high affinity binding of the LxCxE motif to the pRb pocket B subdomain, enabling CR3 to make contact with both the C-terminal domain of pRb and the MB domain of E2F. This concerted effort between both the disordered and structured domains of E7 is necessary for full E2F displacement [31,68,89,105,106].

As an IDP, the overall structural flexibility of E7 enables it to form complexes with a variety of cellular binding partners [103,107,108]. More specifically, E7 has been shown to form a complex with the TAZ2 domain from the transcription co-activator CBP and its homolog p300, creating an interaction that is both high affinity and highly dynamic [109]. Since E7 is considered an obligate dimer within the cellular environment, it has been proposed that dimerization through CR3 provides two disordered regions, each consisting of CR1/CR2, available to bind multiple host cell proteins [103]. This proposed mechanism can account for the facilitated interactions between cellular binding partners that would not otherwise occur and may contribute in part to the targeted degradation of pRb by E7 [103,110].

In addition to the LxCxE motif, CR2 also contains two conserved serine residues (Figure 2), shown to be phosphorylated by CK2, and necessary for optimal function within host cells [63,110,111,112]. E7 from high-risk genotypes, such as HPV16 and HPV18, have a canonical recognition sequence for the kinase CK2, consisting of an uninterrupted sequence of negatively charged amino acid residues (aspartic acid and glutamic acid) in the −1 to +5 positions relative to the phosphoacceptor serine residues [74]. The phosphorylation conducted by CK2 increases binding affinity for pRb as well as other cellular proteins [110,112,113]. Figure 2 shows the relative locations of the LxCxE motif and neighboring phosphoacceptor serine residues. There is some propensity for the formation of transient secondary structural elements within the CR1/CR2 domains, also predicted by PONDR analysis, as shown in Figure 2B,C [114]. However, this entire domain has been experimentally determined to be highly flexible [102,115,116]. This may indicate that the LxCxE motif itself prefers a certain level of transient structural stability, but the regions surrounding it require flexibility. Additionally, since the phosphoacceptor serine residues are within a disordered region already containing negative charges, the addition of negatively charged phosphate groups would further extend the amino acid residues in this highly dynamic domain [117]. Because so much structural information exists for E7 and its relationship to pRb, as well as other cellular targets, this represents an excellent example of a viral oncoprotein both displacing E2F from pRb and targeting pRb for degradation, utilizing an LxCxE motif. Figure 2B,C illustrate the location of the LxCxE motif relative to regions of intrinsic disorder, as well as phosphoacceptor serine residues known to enhance cellular interactions. As such, several of the oncogenic proteins to follow in this review will be directly compared to these patterns seen for high-risk HPV E7.

### 2.2. Polyomaviruses

Polyomaviridae is a family of small, non-enveloped DNA viruses, with icosahedral capsids, measuring approximately 40–50 nm in diameter. Their genomes consist of circular double-stranded DNA (~5 kb), maintained in host cells either as extra-chromosomal episomes or, in some cases of cancer, integrated into the host genome [118,119]. Currently, 15 human polyomaviruses (PyVs) have been identified, with several implicated in tumor progression [120]. In particular, Merkel cell polyomavirus (MCPyV, Alphapolyomavirus quintihominis) and BK polyomavirus (BKPyV, Betapolyomavirus hominis) have been associated with various malignancies [3,4,5].

### 2.3. Merkel Cell Polyomavirus (MCPyV) LTAg Protein Directly Binds and Deactivates pRb Through LxCxE Motif

Merkel cell polyomavirus (MCPyV) was first identified in 2008 from tissue samples of patients diagnosed with Merkel cell carcinoma (MCC), a rare but aggressive neuroendocrine skin cancer [121,122]. Seroprevalence studies indicate that MCPyV infection is widespread, with approximately 60–80% of adults testing positive for antibodies [123]. The virus establishes a lifelong persistent infection in the skin, typically remaining asymptomatic in immunocompetent individuals. However, under conditions of immunosuppression, such as in organ transplant recipients or individuals infected with HIV, MCPyV can integrate into the host genome and drive the oncogenesis of MCC [121,122]. The incidence of MCC in the United States has been rising in recent decades and is predicted to continue to rise as the overall population ages [124,125]. Currently, there are no specific antiviral treatments for MCPyV; however, there are several clinical trials underway [126]. Current MCC management primarily involves surgical excision, radiation therapy, chemotherapy, or immunotherapy [127]. In the case of transplant recipients, the strong link between immunosuppression and MCC requires careful modulation of immunosuppressive therapy to balance the risk of tumor progression against the potential for transplant rejection [124].

The oncogenic process begins with the clonal integration of MCPyV DNA into the host genome, which occurs in approximately 80% of MCC cases, suggesting a crucial role in tumor initiation [121]. The evidence supporting MCPyV as a direct oncogenic driver is strengthened by its clonal integration into MCC tumor cells, demonstrating that viral infection occurs prior to malignant transformation [121]. Furthermore, transgenic mouse models expressing MCPyV develop MCC-like tumors, confirming their oncogenic potential [128]. Both the large tumor antigen (LTAg) and small tumor antigen (sTAg) proteins are essential oncoproteins for viral replication and cancer cell proliferation [129,130]. Viral integration is associated with mutations that introduce premature stop codons truncating the C-terminus of LTAg (Figure 3A), disabling its ability to support viral replication while preserving its capacity to disrupt tumor suppressor pathways [131]. The most common mutations within LTAg for patients with MCPyV-positive MCC are found within the helicase domain, origin-binding domain, and the nuclear localization signal (Figure 3A) [131,132]. Interestingly, the N-terminal domain up to amino acid 252, which includes the LxCxE motif, is maintained in MCC tumors, confirming its fundamental role in MCC progression [119,133]. While LTAg hijacks many cellular pathways, a primary mechanism of oncogenesis involves the disruption of host tumor suppressor pathways, such as pRb/E2F [134]. Like most known polyomaviruses, LTAg from MCPyV contains an LxCxE motif that binds directly to pRb, allowing it to disrupt the pRb/E2F complex and drive the expression of genes required for S-phase entry, resulting in unchecked cell division [22,52,53,119,135,136]. For example, recent studies using transgenic mice, designed with attenuated LTAg-pRb LxCxE-dependent interactions, demonstrated significantly less epithelial hyperplasia and E2F-dependent gene expression [22].

The LxCxE motif itself is located between the MCPyV unique regions (MUR) 1 and 2 (Figure 3B) [130]. We have overlaid the domains of this protein with results from PONDR analysis, suggesting which regions of the protein display intrinsic disorder. The results are similar to those obtained previously, and suggest that the LxCxE motif is located partially adjacent to, and partially within, an IDR [130]. Interestingly, these results are very similar to those of E7 from high-risk HPV, in that amino acids near this binding motif have some small propensity for secondary structure within the context of an IDR (Figure 3B,C). There is strong support for IDPs/IDRs to achieve stable secondary structure in the form of loops and small helices upon binding to a partner protein [137]. Since the LxCxE motif for LTAg is the primary binding domain for pRb, it would make sense that the neighboring amino acid residues would form some level of secondary structure upon stable interaction with pRb. Therefore, once again, we see the appearance of the LxCxE motif in a region of overall intrinsic disorder, with a certain level of predicted transient secondary structure in the vicinity.

Although MCPyV LTAg contains 82 predicted phosphorylation sites, some of which have been directly linked to optimal protein function, the roles of most remain unclear [69,138,139]. Phosphorylation of serine 220, near the LxCxE motif and within an IDR, has been shown to enhance pRb binding and promote cell proliferation [70]. Figure 3B,C show the proximity between the LxCxE motif and serine 220 in the context of predicted disorder within the protein. Unlike HPV E7, serine 220 does not have the canonical substrate recognition sequence for phosphorylation by CK2, consisting of acidic residues in the +1 and +3 positions relative to the phosphoacceptor serine [74]. However, it has been speculated that if the adjacent serine at position 219 is phosphorylated, this would create a negative charge at the -1 position, which could possibly enable hierarchical phosphorylation, which has been previously observed by CK2 [140,141]. However, the presence of proline residues makes this hypothesis challenging [70]. Regardless, it has been shown that phosphorylation at this site is crucial for optimal pRb-induced cell cycle dysregulation [70]. The proximity of serine 220 to the LxCxE motif in the LTAg protein from MCPyV is similar to E7, where we see the presence of a phosphoacceptor serine within several residues of the LxCxE motif within an IDR. However, the broader phosphorylation landscape of LTAg is not well understood, and further studies are needed to clarify how these modifications influence viral oncogenesis and protein function.

### 2.4. BK Polyomavirus (BKPyV) LTAg Protein Directly Binds and Deactivates pRb Through LxCxE Motif

BK polyomavirus (BKPyV) was first identified in 1971 from an immunosuppressed kidney transplant patient [142]. Seroprevalence studies indicate that approximately 79% of young adults globally test positive for BKPyV, reflecting widespread exposure during early life [143,144]. The virus induces a persistent latent infection in the kidney and urinary tract, remaining asymptomatic in immunocompetent individuals. However, under immunosuppressive conditions, such as in kidney or bone marrow transplant recipients, the latent virus can reactivate, leading to significant clinical complications [145,146]. This reactivation of BKPyV is commonly observed in immunosuppressed transplant patients, where it can contribute to BK polyomavirus-associated nephropathy (BKVAN) in kidney recipients and hemorrhagic cystitis in hematopoietic stem cell transplant recipients [147,148]. To mitigate this, the modulation of the immunosuppressants helps patients fight the BKPyV infection, but they then face an increased likelihood of transplant rejection [149,150].

BKPyV not only plays a role in certain pathologies such as BKVAN and hemorrhagic cystitis but has likewise been found to be associated with urologic tumors in patients undergoing immunosuppressive therapy [5,142]. Studies have reported an 83% increased risk of malignancy in transplant patients with BKPyV-induced nephropathy compared to transplant patients without elevated viral replication [151]. Furthermore, numerous case studies have linked active BKPyV infections, as evidenced by the expression of BKPyV genes, within renal and urinary tract tumors, including bladder, kidney, ureter, and pelvic carcinomas [152]. Most convincingly, for multiple transplant patients undergoing immunosuppressive therapy, in addition to viral episomes, BKPyV was found clonally integrated in the host genome of kidney carcinomas and urothelial carcinomas [153,154,155,156]. The viral integration of BKPyV genes into the host genome supports a causal effect, as opposed to merely an associative role of BKPyV in driving tumor progression.

Like MCPyV, expression of the two viral oncogenes LTAg and sTAg from BKPyV are thought to play key roles in cellular transformation [51]. LTAg drives cell proliferation by interacting with the host’s tumor suppressors, p53 and pRb. In contrast, sTAg inhibits protein phosphatase 2A and stimulates the MAP kinase pathway, likewise causing uncontrolled proliferation. Studies suggest that aggressive malignancies are more commonly associated with LTAg integration into the host genome, reinforcing the hypothesis that viral integration contributes to oncogenic transformation [154,156]. This integration allows for upregulation of LTAg which, in turn, disrupts the tumor suppressive activities of p53 and pRb and drives proliferation culminating in oncogenesis.

Following previously observed patterns, LTAg from BKPyV binds directly to pRb through the LxCxE motif [157]. Figure 4 shows this binding motif located in the linker region between the DNAJ domain and the origin-binding domain (OBD), both of which are crucial for viral replication [158,159]. Previous work has proposed a “chaperone” model for E2F displacement, whereby LTAg binds the pRb/E2F complex, but E2F displacement is driven by the recruitment of heat shock cognate 71 kDa protein (HSC70) through the DNAJ domain at the N-terminus [160,161]. It is believed that LTAg therefore requires multiple domains to disrupt the pRb/E2F complex and the flexibility between domains enables additional interactions within the complex [159]. Looking at Figure 4, the LxCxE motif is located in a region immediately adjacent to predicted intrinsic disorder. This follows the same pattern seen for the other viral oncoproteins provided and supports the idea that flexibility appears crucial to mediating E2F displacement from pRb. Phosphorylation of LTAg from BKPyV has been observed, but little is currently known about the specific mechanism [65]. Serine 114, located adjacent to the LxCxE motif (amino acid residues 105–109), has been identified as a phosphoacceptor residue and follows the pattern seen for the other pRb-binding proteins (Figure 4). While there is little information regarding the kinase involved in the phosphorylation of serine 114, analysis of the amino acid sequence reveals the presence of aspartic and glutamic acid residues in the +1, +2, and +3 positions relative to the phosphoacceptor, which is highly indicative of a recognition sequence for CK2 [74]. While additional investigation is necessary, it appears that LTAg from BKPyV is following the pattern observed for the other oncogenic proteins, which dysregulate the cell cycle through an LxCxE-dependent interaction with pRb.

## 3. LxCxE-Mimic Strategies: Human T-Cell Lymphotropic Virus Type 1, Hepatitis C Virus, and Epstein–Barr Virus

### 3.1. Human T-Cell Lymphotropic Virus Type 1 (HTLV-1) Tax Directly Inhibits pRb Function Through LxCxE-Mimic Motif

Human T-cell lymphotropic virus type 1 (HTLV-1) was the first retrovirus discovered in humans [162]. As a retrovirus, it reverse transcribes its RNA genome upon cell entry to produce DNA that it then integrates into the genome of its host [163,164]. While there are other oncogenic retroviruses, such as mouse mammary tumor virus (MMTV) [165] and human T-cell lymphotropic virus type 2 (HTLV-2), HTLV-1 is currently the only one known to cause cancer in humans. It is estimated that 15 to 20 million people worldwide are infected with HTLV-1, with hotspots in Southwestern Japan, islands throughout the Caribbean, as well as parts of South America, sub-Saharan Africa [166,167], and, more recently noted, remote Indigenous communities within central Australia [168]. Initial infection leads to viral integration into host CD4^+^ T-cells and approximately 5% of cases lead to a highly aggressive subtype of T-cell lymphoma, adult T-cell leukemia/lymphoma (ATL), often occurring decades after the initial infection [169,170,171,172]. During this latent period, infected cells exist as memory CD4^+^ T cells with minimal impact on cellular transcription, enabling the virus to remain undetected by the immune system [173,174,175] while continuing to maintain and replicate infected cells [176]. While the mechanisms for this latency period in HTLV-1 are less known, the onset of ATL has been associated with specific oncogenic driver mutations within host cells [177]. Treatment of ATL is limited, mostly involving chemotherapy, several antiviral therapies, and hematopoietic stem cell transplants [178,179], suggesting that once HTLV-1 has hijacked the cell cycle of its host, the prognosis is poor.

Among the proteins encoded by HTLV-1, Tax is considered one of the dominant oncoproteins, playing a crucial role in viral transcription and host cell transformation [180,181,182]. While Tax has a wide range of host cell targets involved in proliferation, ranging from transcription factors, transcriptional activators, and cell signaling pathways, interference with the pRb/E2F cell cycle regulation system provides a major strategy for HTLV-1 oncogenesis [183]. The presence of Tax was shown to result in the stimulation of the G1 to S-phase transition in the cell cycle of primary T cells, suggesting potential influence over the pRb/E2F regulatory system [184]. Tax has been shown to specifically bind Cyclins-D1, -D2, and -D3, as well as CDK4, and CDK6 through its N terminal domain, but not CDK1 or CDK2 [46]. Interaction between Tax and the Cyclin D/CDK complex appears to enhance the activity of these cyclin-dependent kinases in the phosphorylation and subsequent deactivation of pRb [185]. In the presence of mutant forms of Tax, unable to bind CDK4, this enhanced kinase activity was diminished. These results suggest that Tax interactions with CDK4 result in an increased capability to phosphorylate pRb, ultimately resulting in deactivation of pRb and subsequent accelerated initiation of S phase associated with tumorigenesis [46,183].

Given the accelerated S phase entry and faster growth kinetics observed in cells expressing Tax [186], it stands to reason that this oncoprotein utilizes multiple strategies to dysregulate the pRb/E2F complex. In fact, previous work has shown that, in addition to its increased activation of CDK4, Tax can also upregulate and sequester other proteins involved in cell cycle checkpoints, enabling both increased phosphorylation and degradation of pRb [47,187,188]. These decreased pRb levels in cells expressing Tax suggest that Tax could also target pRb through direct binding [43]. Analysis of the amino acid sequence of Tax identified an LxCxE-mimic domain in its C-terminal region, which can interact directly with pRb and target it for proteasomal degradation [43]. Figure 5 shows the domain sequence for Tax, with the binding domain for Cyclin D/CDK located within the N-terminal NLS domain, as well as the proposed LxCxE-mimic motif at the C-terminus. Similar to the patterns seen for HPV16 E7 and LTAg from both MCPyV and BKPyV, the pRb-binding domain for Tax is located adjacent to an IDR, suggesting the need for structural flexibility near this region. Interestingly, the binding site for Cyclin D/CDK and the proposed pRb-binding site are on opposite ends of the Tax protein (Figure 5). Just as IDRs have been shown to be important for the formation of ternary complexes with pRb and other cellular proteins, ultimately leading to pRb degradation [103], the ability of Tax to influence pRb function through phosphorylation by Cyclin D/CDK could be enhanced through its ability to bind the cyclin complex and pRb simultaneously [103,189]. Finally, there are two adjacent serine residues at amino acid positions 300 and 301 within several residues of the proposed LxCxE-mimic motif. It has been demonstrated that these serine residues are phosphorylated and, while the kinase is currently unknown [67], this suggests additional similarities to proteins that interact with pRb in an LxCxE-dependent manner. Ultimately, further investigations into these interactions would be necessary to fully understand this complex system.

### 3.2. Hepatitis C Virus (HCV) NS5B Directly Binds and Inhibits pRb Function Through LxCxE-Mimic Motif

The hepatitis C virus (HCV) is part of the Flaviviridae family, which are small (measuring 40 nm in diameter), enveloped, icosahedral, positive single-stranded RNA (+ssRNA) viruses. Worldwide, 1.0–2.5% of the population has an HCV infection, varying greatly depending on geography, with the highest incidence rates in the Eastern Mediterranean and European regions [192]. Of those infected, 30% will naturally clear the acute infection on their own. For the remaining 70%, approximately 71 million people, the chronic infection can last decades [193]. HCV is a blood-borne pathogen, readily spread through the improper sterilization of needles and unscreened blood transfusions. The identification and characterization of HCV took decades, until 1989, when the virus was isolated as a result of the work by Michael Houghton, Harvey Atler, and Charles Rice, for which they were awarded the Nobel Prize in 2020 [194]. For patients with latent infections, this chronic inflammation can lead to hepatic fibrosis, cirrhosis (scarring and damage in the liver), and hepatocellular carcinoma (HCC), which can cause increased morbidity and mortality (290,000 deaths annually worldwide) [195]. The HCV-induced transition from hepatitis to cirrhosis to HCC can take decades with the progression towards cancer further exacerbated by comorbidities, such as alcohol consumption, hepatic steatosis, and diabetes. Further complicating our understanding of this disease is the fact that there are numerous HCV genotypes that differentially impact disease progression. Currently, there are six major genotypes and numerous subtypes [196,197]. Approximately half of the HCCs in the United States are attributable to sustained HCV infection, most often associated with HCV subtypes 1a and 1b [198].

The 9.6 kb RNA genome of HCV is transcribed as a single transcript within the cytoplasm that is later spliced into 10 viral proteins, some of which are structural (such as the core and envelope proteins), and others that facilitate viral replication. In contrast to the DNA viruses previously discussed, HCV is an RNA virus that does not translocate to the nucleus or integrate within the host genome. Since progression towards carcinogenesis takes decades, it is challenging to delineate the profound role of chronic inflammation due to HCV infection from the direct role of particular viral proteins on the host’s cellular machinery. There is evidence that the core protein, as well as several non-structural (NS) proteins (NS3, NS5A, and NS5B), play a role in the dysregulation of the cell cycle. One key oncoprotein, NS5B, an RNA-dependent RNA polymerase, has been shown to directly bind to pRb utilizing an LxCxD (LVCGD) motif (amino acids 314–318), which is a mimic of the canonical LxCxE [54]. The pRb/NS5B complex then recruits the host’s E6-associated protein (E6AP), which ubiquitinates pRb and targets it for proteasomal degradation, further driving uncontrolled cell division [44,199].

The structure of NS5B, resembling the shape of a right hand, was determined in 1999 and involves three subdomains labeled finger, thumb, and palm, which surround the active site [200,201,202]. Analysis of the NS5B sequence using PONDR reveals a region of disorder (amino acids 326–365) immediately adjacent to the LxCxD motif within the thumb domain (amino acids 314–318). Despite most of the protein containing secondary structure, this is one of the few IDRs within the sequence. Looking at Figure 6, this is consistent with the LxCxE-dependent oncoviruses shown in Figure 2, Figure 3 and Figure 4. Furthermore, NSB5 has a phosphoacceptor serine residue at position 326, which is within 8 amino acid residues of the LxCxD motif. While this residue is phosphorylated by Akt kinase and not by CK2, it has been shown that phosphorylation at this position is necessary for optimal protein activity [75]. While a connection has not yet been made between phosphorylation and pRb affinity or activity, the location of this phosphoacceptor serine fits the previously observed pattern and presents an interesting opportunity for further studies.

### 3.3. Epstein–Barr Virus (EBV) EBNA-3C Directly Binds pRb Through a Potential LxCxE-Mimic Domain

The Epstein–Barr virus (EBV), also known as human herpesvirus 4 (HHV-4), is a double-stranded DNA virus measuring 120–180 nm in diameter and belongs to the gammaherpesvirus family [205,206]. EBV was the first virus to be identified in a human tumor of Burkitt Lymphoma in 1964 [207]. It is one of the most common viruses in humans and it is estimated that about 95% of the world population has asymptomatic infection with EBV [208,209]. Approximately 250,000 cases of cancer globally are attributable to EBV infection [80,210]. In the United States, 90% of adults show evidence of past EBV infections [211]. While it was originally shown to primarily infect B-cells, it is also capable of infecting epithelial cells and has been implicated in a variety of human cancers, including Hodgkin’s lymphoma, several subtypes of non-Hodgkin’s lymphoma, T-cell lymphoma, NK/T cell lymphoma, nasopharyngeal carcinoma, and gastric cancers [212,213,214]. Additionally, coinfection of EBV with HPV can lead to the progression of several cancers through increased DNA integration [215,216]. Primary infection can occur at any point in life, but infection occurring after childhood often manifests as infectious mononucleosis [217]. Like other herpesviruses, EBV has both latent and lytic phases [218]. During the latent phase, it resides in memory B-cells and avoids host immunity while relying on normal cellular division to set up a predominantly asymptomatic lifelong infection [219]. This latency phase can undergo one of three lifecycles, Latency Phase I, II, or III, each of which leads to a distinct set of viral proteins [220]. Under certain conditions, the virus will reactivate and enter the lytic phase where all viral genes are expressed, and the virus can be released to reinfect cells of the same host or be transmitted to a new host [221].

While the latency phase is often associated with oncogenesis, the lytic phase also plays a role in EBV-associated cancers. During latency, the EBV genome encodes a variety of nuclear antigens (EBNAs) and membrane proteins (LMPs), all of which are necessary to maintain constant viral presence throughout the lifespan of the host [222]. EBV is highly proficient in increasing the proliferation and immortalization of B-cells [223]. Viral progression to tumor formation is most often associated with the expression of Latent Membrane Protein 1 (LMP1), EBV nuclear antigen LP (EBNA-LP), EBV nuclear antigen 2 (EBNA-2), and EBV nuclear antigen 3 (EBNA-3) [224,225,226]. EBNA-3 is composed of three closely related proteins, -3A, -3B, and -3C, with -3A and -3C essential for B-cell transformation [227]. Specifically, the presence of EBNA-3C in cells was shown to increase the hyperphosphorylation and inactivation of pRb as well as the transcriptional activity of E2F, suggesting that EBNA-3C has a direct role in the G1-S transition in cells by targeting the pRb/E2F complex [45,228].

The protein EBNA-3C is essential for viral transformation of host cells and has been shown to dysregulate the pRb/E2F complex through both indirect and, more recently, direct interaction with pRb [45,228,229,230,231]. Indirectly, EBNA-3C is linked to increased activity of Cyclin/CDK complexes, which contribute to the phosphorylation and subsequent inactivation of pRb [230]. It has been shown previously that EBNA-3C can enhance the kinase activity of CDK1/Cyclin D6, which enables increased phosphorylation of pRb [227]. Additionally, EBNA-3C was shown to recruit the Skp1Cul1F-box complex, the SCF^Skp2^ ubiquitin ligase complex. This complex is known to regulate the stability of E2F, but recruitment by EBNA-3C was shown to result in the ubiquitination and subsequent degradation of pRb, thus promoting the transition to S phase [230]. This particular study also demonstrated co-immunoprecipitation between EBNA-3C and pRb, thus revealing that EBNA-3C was directly binding pRb and targeting it for degradation [230]. Further studies revealed that the pRb-binding interface was occurring through the N-terminal region of EBNA-3C [230]. A comparison to HPV E7 shows some sequence homology near the LxCxE region and additional analysis led to the conclusion that EBNA-3C binds to pRb primarily through amino acids 141–145, which, based on alignments with E7 from HPV16, includes a potential LxCxE-mimic domain (ILCFV) [230]. Figure 7 shows this region, along with the entire sequence, mapped to the results from PONDR analysis. Unlike the proteins previously discussed, the pRb-binding domain is located within a region predicted to have secondary structure, which differs from the oncoproteins seen previously. However, there are large regions of predicted disorder throughout, with PONDR calculating 66.8% overall intrinsic disorder, suggesting significant structural flexibility. Additionally, NetPhos 3.1 predicts that the serine residue at position 134 is a phosphoacceptor and lists CK2 as a potential kinase. This would put a CK2 substrate phosphoacceptor serine within several residues of the LxCxE-mimic region, which conforms to several of the patterns shown throughout this review. Interestingly, unlike the other oncogenic proteins in this study, EBNA-3C does not appear to be involved in the degradation of p107 or p130 pocket proteins but instead appears specific to pRb [230]. Hopefully, some of these patterns can help to guide future studies of the mechanistic details of these interactions.

## 4. LxCxE-Independent or Indirect pRb Interaction: Kaposi Sarcoma-Associated Herpesvirus and Hepatitis B Virus

### 4.1. Kaposi Sarcoma-Associated Herpesvirus (KSHV) LANA Directly Binds pRb Through Its C-Terminal Region

Kaposi sarcoma-associated herpesvirus (KSHV), or human herpesvirus 8 (HHV-8), is an enveloped, double-stranded DNA virus measuring approximately 150–200 nm in diameter [232,233]. KSHV is considered a gammaherpesvirus due to its double-stranded linear DNA, measuring approximately 165–170 kb [234]. KSHV infection can lead to the development of Kaposi sarcoma (KS) [235]. KS can be divided into four categories: classic KS, endemic (African) KS, epidemic (AIDS-associated) KS, and iatrogenic (transplant-related) KS [235]. Although these are unique strains, all originate from KSHV infection. Additionally, KSHV is linked to both primary effusion lymphoma (PEL) and multicentric Castleman’s disease (MCD) [234,236].

The seroprevalence of KSHV varies geographically with KSHV most prevalent in African populations, having 50% seroprevalence. Mediterranean countries have 10–25% seroprevalence, which decreases to less than 10% in Europe, Asia, and the United States [234]. Evidence suggests saliva as the primary means of transmission, in addition to sexual contact, blood transfusions, and transplants [235,237]. KSHV infects an array of cells and has both a latent cycle (default lifecycle) and a lytic cycle. During the latent cycle, KSHV DNA is tethered to the host’s genome to ensure the passage of viral information onto the host’s daughter cells during mitosis. In the lytic cycle, new viral capsids are expressed and released, destroying the host cell in the process [234]. The development of KS requires the host to have an infection with KSHV combined with immune impairment. KSHV infection leads to lesions with three developmental stages: the patch stage, as macules, which then evolve into plaques, and, finally, present as larger nodules once oncogenic. The tumors can ulcerate, cause swelling, grow in extracutaneous regions, or invade other tissues. Currently, there is no definitive cure for KS. Treatments to slow disease progression include the removal of lesions via surgery for cosmetic comfort or tumor control purposes, or other cytotoxic therapies [238]. Additionally, primary effusion lymphoma (PEL) is linked to KSHV infection. Like KS, PEL has a higher likelihood of manifestation with the combination of immunodeficiencies. PEL can result in effusion buildup, which carry cancerous cells, around the pleural, peritoneal, or pericardial space. Treatment for PEL can include chemotherapy, antiviral drugs, and, in rare cases, stem cell transplants [239].

The oncogenic manifestation of KSHV requires dysregulation of the cell cycle through both indirect and direct manipulation of the pRb/E2F complex. Two different viral genes have been identified as having oncogenic potential: open reading frame 73 (ORF73), which encodes the latency-associated nuclear antigen (LANA) [240], and open reading frame 72 (ORF72), which expresses a Cyclin D-like homolog [239]. As discussed, the Cyclin D/CDK complex phosphorylates pRb, which results in the initiation of the cell cycle [241]. Multiple studies have suggested that KSHV-cyc (Cylin D-homolog) operates as a cyclin protein by stimulating CDK6. The CDK6-KSHV-cyc complex can then go on to phosphorylate and deactivate pRb, which may lead to cell cycle progression [242,243].

The protein LANA is known for its role in tethering viral DNA to the host cell’s chromosomes, ensuring the viral genome will be passed to progeny cells during mitosis [244]. LANA is a relatively large protein containing predominantly basic amino acid residues in the N- and C-terminal domains and a hydrophilic central domain containing mostly polar/charged residues [233]. It was shown that LANA successfully transactivated regions of an artificial promoter containing E2F-binding sites, suggesting that LANA can act as a cofactor in gene expression that is E2F dependent [191]. Additionally, LANA was found to upregulate the CCNE1 promoter, a gene regulated by both E2F and members of the pRb protein family, and that is necessary for cell cycle progression [191,245]. It was also demonstrated that LANA directly binds to the hypophosphorylated form of pRb through its C-terminal region (Figure 8) [191,246]. To explore whether this interaction had oncogenic potential, the morphological state of Saos2 cells was studied. Typically, pRb leads to the growth arrest of Saos2 cells [247]. However, introducing LANA-1 proved to reverse this state, suggesting its role in mediating cell cycle stages through pRb interactions [191].

Limited structural information is currently available for LANA, but looking at the predicted secondary structure by PONDR analysis (Figure 8A,B), the majority of LANA is predicted to be disordered, with pRb interaction confirmed to be occurring in a region between amino acid residues 809 and 990. This highly dynamic region contains predominantly leucine, glutamic acid, and glutamine residues [248]. While they have not identified an LxCxE motif in LANA, there are many homologous residues to HPV16 E7 within the pRb-binding domain. Additionally, within the pRb-binding region of LANA, there is one threonine residue at position 899 and four serine residues at positions 909, 910, 911, and 923, all of which NetPhos3.1 predicted to be phosphorylated by CK2. These serine residues are labeled in Figure 8. Further investigation into the phosphoacceptor serine residues, combined with potential LxCxE-mimic domains, could help elucidate the mechanistic details of the pRb-binding interface for this protein.

### 4.2. Hepatitis B Virus (HBV) HBx Indirectly Inhibits pRb Activity with the Potential for Direct Interaction

Hepatitis B virus (HBV) is part of the *Hepadnaviridae* family, which are small (measuring 42 nm in diameter), enveloped, icosahedral, partially dsDNA viruses. Worldwide, approximately 3.5% of the population has an HBV infection, with prevalence varying significantly by region, particularly in sub-Saharan Africa and East Asia [250]. While over 90% of adults infected with HBV naturally clear the acute infection, around 10% develop chronic hepatitis B, leading to long-term liver complications such as cirrhosis and HCC [251]. Chronic HBV infection affects an estimated 296 million people globally, contributing to nearly 820,000 annual deaths due to liver disease and complications [252]. HBV is a blood-borne pathogen that is highly transmissible through perinatal transmission, unprotected sexual contact, sharing of needles, and exposure to infected blood products. The identification of HBV dates back to 1963 when Baruch Blumberg discovered the Australia antigen (HBsAg), leading to the development of the first hepatitis B vaccine. For his pioneering work, Blumberg was awarded the Nobel Prize in Physiology or Medicine in 1976 [253].

For patients with chronic HBV infections, persistent hepatic inflammation can lead to hepatic fibrosis, cirrhosis, and HCC, significantly increasing morbidity and mortality [249]. While the progression from chronic hepatitis to cirrhosis and, ultimately, to HCC can take decades, the risk is exacerbated by comorbidities such as alcohol consumption, metabolic dysfunction-associated steatotic liver disease (MASLD), and diabetes [253,254,255,256]. Additionally, disease severity varies depending on the HBV genotype. There are ten major HBV genotypes (A–J), with differences in disease progression and response to antiviral therapy [257]. Approximately 54% of HCC cases globally are due to HBV infection, with China experiencing the highest incidence rate [258]. The 3.2 kb partially double-stranded DNA genome of HBV is transported to the nucleus, where it is converted into covalently closed circular DNA (cccDNA) that serves as a template for viral transcription. This transcription produces multiple viral RNAs, including a 3.5 kb pregenomic RNA (pgRNA), which is reverse-transcribed by the viral polymerase into new viral DNA within the nucleocapsid. HBV can integrate portions of its DNA into the host genome, leading to genomic instability and insertional mutagenesis [259]. The HBV X protein (HBx), a non-structural regulatory protein, further contributes to oncogenesis by modulating cellular signaling pathways, disrupting tumor suppressor mechanisms, and altering epigenetic regulation [48].

HBx is known to interact with various cellular pathways to promote tumorigenesis. For example, it was shown to interfere with the tumor suppressor protein p53 by inhibiting its DNA repair functions and promoting overall genomic instability [260]. Additionally, HBx can bind to damaged DNA-binding protein 1 (DDB1), impacting the ubiquitin-proteasome pathway and disrupting cell cycle regulation. This leads to uncontrolled cell proliferation and survival [261]. HBx is known to interfere with the pRb tumor suppressor pathway, primarily by modulating the activity of E2F transcription factors. For example, HBx has been shown to increase CDK2 activity, leading to alterations in the E2F-Rb balance and favoring the expression of genes involved in cell cycle progression, such as CDC6 [262]. While the structure of HBx is not entirely known, it has been shown to have IDRs. The results from PONDR analysis shown in Figure 9 suggest several regions of intrinsic disorder, along with some structural domains, which agree with previous studies [263]. The structure includes three conserved regions with the N-terminal domain predicted to be largely unstructured [264]. The full-length version of this protein has been difficult to study from a structural standpoint, due to its dynamic nature and difficulty in recombinantly producing sufficient quantities for structural analysis. HBx is also predicted to form homodimers via disulfide bonds and has been described as having high levels of conformational flexibility [265]. Overall, much of the structural information for HBx remains unknown.

While it has not currently been demonstrated that HBx binds directly to pRb [266], structural patterns exist within the protein sequence when compared to the other oncogenic proteins in this review, which suggests the potential for a direct interaction. Looking at Figure 9A,B, PONDR scores combined with NetPhos3.1 analysis revealed a known phosphoacceptor serine residue at position 25, within an IDR, which was predicted to be a substrate for CK1 [267]. Further alignment of the HBx sequence with that of E7 from HPV16 revealed some homology with the CR1/CR2 domains of E7. Figure 9C shows Clustal Omega multiple sequence alignment with highlighted similarities within the LxCxE region of E7, suggesting that HBx may contain an LxCxE-mimic domain similar to that of Tax from HTLV-1 [43]. Additionally, serine 25 aligned with one of the phosphoacceptor serines of E7 known to increase the affinity for pRb [68]. While there is no current information on direct interaction between HBx and pRb, this not only represents an area for future investigation, but also demonstrates the importance of comparing similarities between these viral oncogenic proteins as a way to investigate potentially novel binding partners within cells.

## 5. Conclusions

### 5.1. Viruses and Cancer Biology

The heterogeneous nature of cancer presents one of the greatest medical challenges of the current age. However, some cancers due to infectious agents are theoretically preventable. A mechanistic understanding of these oncogenic viruses, and the common means by which they manipulate the host cell, not only aids in prevention and treatment, but also provides a unique glimpse into a better understanding of cancer biology. This review looks at eight oncogenic viruses, all of which ultimately induce dysregulation of the cell cycle. Although most of the viruses discussed do this through a multi-pronged approach, they all converge on pRb as a key regulator of this process. Each of these viruses has at least one oncoprotein, which modifies the activity of this key tumor suppressor (Table 1). Interestingly, we observed that these viral oncoproteins tend to have a high degree of intrinsic disorder, imparting a level of flexibility that appears to aid in their manipulation of pRb. The connection previously observed between intrinsic disorder and post-translational modifications is apparent here in that many of these oncogenic viral proteins display an increased ability to bind and deactivate pRb when they are phosphorylated. Phosphorylation additionally requires the use of host cell kinase enzymes, and the ubiquitously expressed and constitutively active CK2 appears to be often hijacked for this purpose. Recognizing patterns such as LxCxE/LxCxE-mimic domains associated with IDRs and phosphorylation by specific host kinases not only helps in further understanding the proteins described here but can provide additional tools in the identification of new oncogenic viral proteins.

### 5.2. Viral Detection and Prevention

The identification of such molecular signatures will become critical as we continue to discover new viruses with oncogenic potential. In the 1990s, there were approximately 120 known genotypes of HPV. Twenty-five years later, more than 200 genotypes have been identified [80]. As such, we presume that there are additional viruses, beyond the eight discussed here, each potentially having multiple genotypes, which play a role in oncogenesis. Effective identification of such novel viruses associated with cancer benefits from a targeted approach. Studying immunocompromised individuals who succumb to cancer due to their inability to fight off common infections may be a good indicator of transmissible agents linked to tumorigenesis [2]. In terms of prevention, vaccines currently exist for HBV and HPV, which appear to be a highly effective strategy for the prevention of cancer caused by viruses. For example, the hepatitis B vaccine has been shown to provide over 90% protection against infection, thereby significantly reducing the risk of chronic liver disease and HCC [268]. Widespread vaccination programs have thus led to a dramatic decline in HBV prevalence, particularly in countries that have implemented universal immunization policies [269]. Additionally, the HPV vaccine in various forms (Gardasil©, Gardasil 9©, and Cervarix©) has been available since 2006 and is targeted toward adolescents and young adults [270]. To fully ascertain its effectiveness in terms of cancer prevention will take decades. However, early indicators of efficacy include a decreased incidence of genital warts, and the nearly complete prevention of latent infections and precancerous lesions associated with several high-risk HPV genotypes, including HPV16 and HPV18 [271,272]. In the United States, women younger than 25 years of age who have been vaccinated have a 65% reduction in cervical cancer looking at incidence rates between 2012 and 2019 [273]. Such hope-filled studies should help drive future research intent on preventing cancer as much as treating it.

## 6. Materials and Methods

### 6.1. PONDR^®^ VLXT

Intrinsic disorder distributions were determined for each of the oncoproteins listed in Table 1 using the bioinformatics tool PONDR^®^ VLXT (Predictor of Natural Disordered Regions). The outputs range from 0 to 1, where below 0.5 is considered ordered and above 0.5 is considered disordered. The characterization as disordered applies to the backbone of the polypeptide chain, as opposed to its side chains.

### 6.2. NETPHOS-3.1

NetPhos 3.1 is a bioinformatics tool that predicts the phosphorylation of serine, threonine, or tyrosine amino acids. In addition, predictions are made as to the likelihood of specific kinases facilitating phosphorylation at a particular amino acid. The kinases include ATM, CKI, CKII, CaM-II, DNAPK, EGFR, GSK3, INSR, PKA, PKB, PKC, PKG, RSK, SRC, cdc2, cdk5, and p38MAPK.

### 6.3. Protein Sequence Alignment

The alignment of HBx from HBV with the disordered domain of E7 from high-risk HPV16 was performed using Clustal Omega multiple sequence alignment from the EMBL-EBI Job Dispatcher sequence analysis tools framework (https://www.ebi.ac.uk/jdispatcher/msa/clustalo, accessed on 23 April 2025) [274].

### 6.4. ACCESSION Numbers 

The NCBI accession numbers for the viral oncoproteins utilized are included here in Table 2.

## Figures and Tables

**Figure 1 viruses-17-00835-f001:**
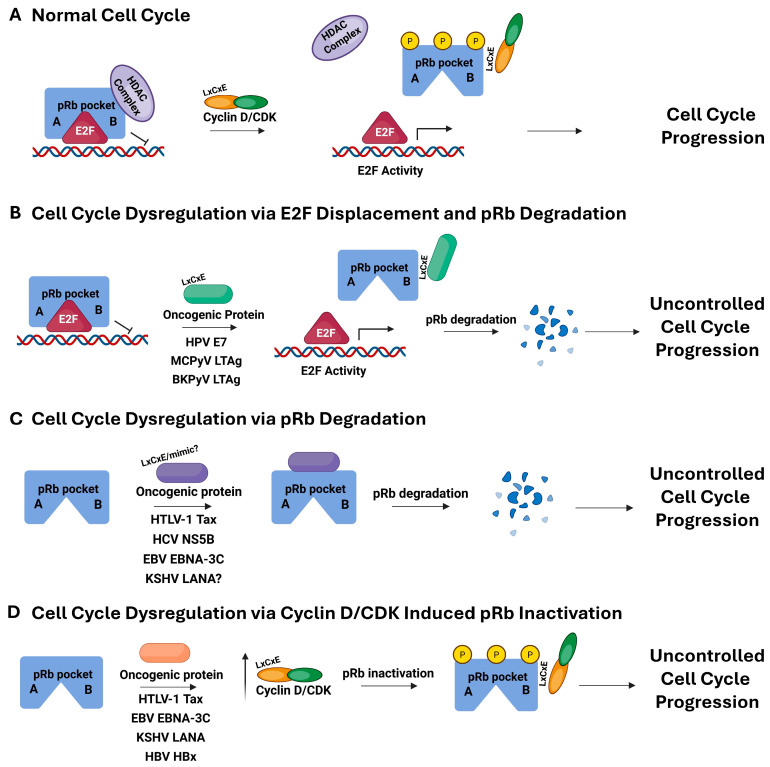
Schematic diagram of pRb in the process of cell cycle progression for normal cells, with the A/B pocket region of pRb represented in blue with the TAD of E2F shown as a red trianble. Phosphorylation by Cyclin D/CDK is represented in yellow on pRb (**A**), cells dysregulated by oncogenic viral proteins known to bind pRb, causing displacement of E2F and degradation of pRb (**B**), pRb targeted degradation, but no known impact on E2F displacement (**C**), and indirect pRb inactivation through upregulation of the Cyclin D/CDK complex (**D**). The oncogenic viruses and their corresponding oncoprotein known to be responsible for these uncontrolled cell cycle progression processes are listed under the arrows labeled oncogenic protein. This figure was generated using Biorender (https://BioRender.com, accessed on 23 April 2025).

**Figure 2 viruses-17-00835-f002:**
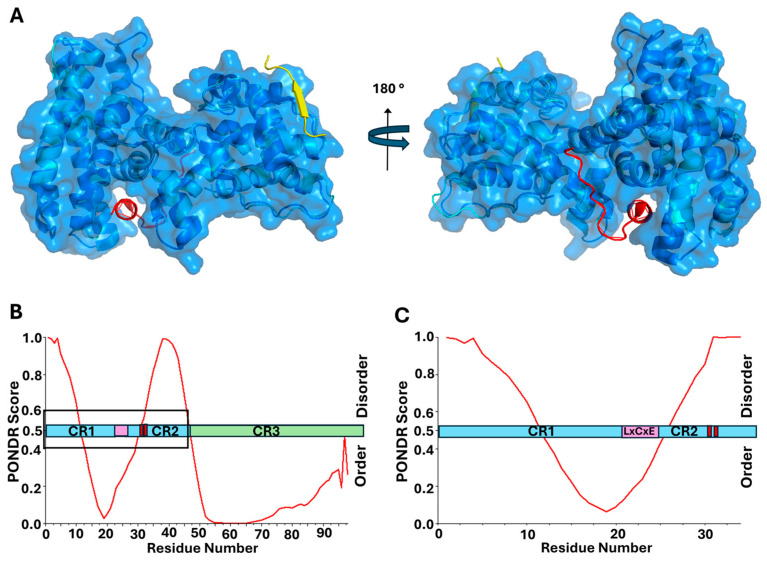
LxCxE-dependent oncoprotein E7 from HPV16 utilizes intrinsic disorder and phosphorylation in conjunction with pRb interaction. (**A**) Structural alignment, performed using PyMOL (PyMOL 3.1, Schrodinger, LLC), of the co-structures for pRb A/B (blue) with the TAD of E2F, adapted from PDB ID 1n4m (red), and the LxCxE motif of E7, adapted from PDB ID 1gux (yellow), show their distinct binding regions. The protein sequence for E7 was overlaid on PONDR scores predicting regions of potential disorder, which correlate to scores greater than 0.5. Results show 27.6% intrinsic disorder for high-risk HPV16 E7, highlighted in light blue. The LxCxE motif is shown in purple and the sites of phosphorylation adjacent to LxCxE are depicted as red lines. (**B**) HPV16 E7, full-length amino acid sequence with the LxCxE motif and serines 31/32 highlighted by a box. (**C**) Zoomed in region within the box highlighted in Figure 2B [98,99,103].

**Figure 3 viruses-17-00835-f003:**
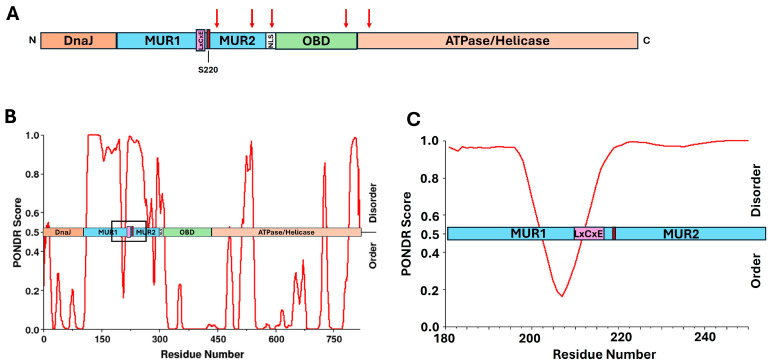
LxCxE-dependent oncoprotein LTAg from MCPyV appears to utilize intrinsic disorder and phosphorylation in conjunction with pRb interaction. (**A**) Schematic diagram of the LTAg from MCPyV shown in detail. The most common sites where mutations from patients with MCC-derived MCPyV are depicted as red arrows. These mutations introduce a premature stop codon truncating the C-terminus. (**B**) The protein sequence for LTAg is overlaid on PONDR scores predicting regions of potential disorder, correlating to scores greater than 0.5, colored in light blue. Known structural domains are labeled with the LxCxE motif colored purple and the phosphoacceptor serine 220 in red. Full-length protein is predicted to be 32.7% disordered with the LxCxE motif and serine 220 highlighted by a box. (**C**) Zoomed in region within the box from Figure 3B.

**Figure 4 viruses-17-00835-f004:**
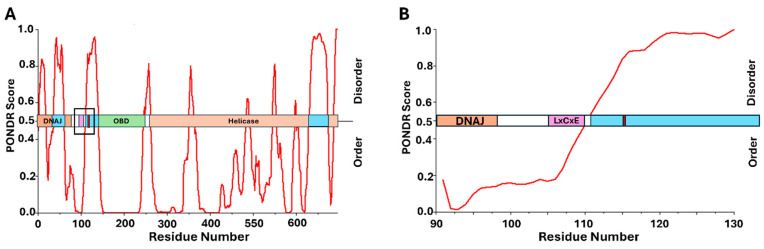
LxCxE-dependent oncoprotein LTAg from BKPyV appears to utilize intrinsic disorder and phosphorylation in conjunction with pRb interaction. Protein sequence for LTAg overlaid on PONDR scores predicting regions of potential disorder, which correlates to scores greater than 0.5. Results show 24.5% predicted disorder, highlighted in light blue. The LxCxE motif is shown in light purple and the site of phosphorylation adjacent to LxCxE is depicted as red lines. (**A**) Full length shows the LxCxE and phosphoacceptor serine 114 residues highlighted by a box. (**B**) Zoomed in region from the box from Figure 4A [119].

**Figure 5 viruses-17-00835-f005:**
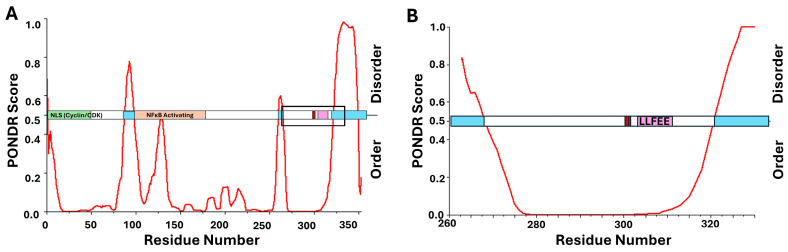
LxCxE-mimic-dependent oncoprotein Tax from HTLV-1 utilizes phosphorylation and disorder in pRb interaction. Known domains for the oncogenic protein Tax from HTLV-1 overlaid on results from PONDR analysis demonstrating regions of potential disorder (PONDR score greater than 0.5). Results demonstrate a 13% disorder for Tax from HTLV-1 (light blue). LxCxE-mimic regions are highlighted in light purple and sites of potential phosphorylation sites are shown in red. (**A**) Full-length Tax from HTLV-1 has several known domains labeled within predominantly structured regions. The potential LxCxE-mimic motif was shown to be in the C-terminal region, highlighted by a box [43,190,191]. (**B**) Zoomed into the region within the box highlighted in Figure 5A.

**Figure 6 viruses-17-00835-f006:**
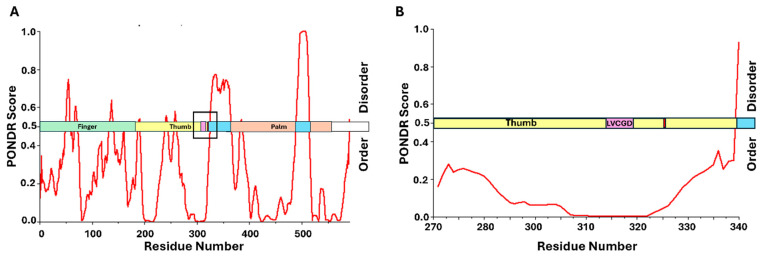
LxCxE-mimic-dependent oncoprotein NS5B from HCV utilizes intrinsic disorder and possibly phosphorylation in pRb interaction. Known domains for the oncogenic protein NS5B from HCV overlaid on results from PONDR analysis demonstrating regions of potential disorder (PONDR score greater than 0.5), with 18.6% predicted disorder (light blue). LxCxE-mimic region is highlighted in purple and site of phosphorylation are shown in red. (**A**) NS5B is a relatively structured protein with several known domains. PONDR predicted a region of disorder adjacent to the LxCxD motif, with the potential phosphoacceptor site at serine 326, highlighted by a box [44,203,204]. (**B**) Zoomed in region within the box highlighted in Figure 6A.

**Figure 7 viruses-17-00835-f007:**
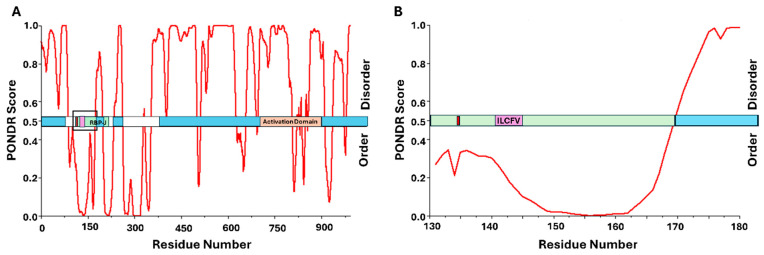
LxCxE-mimic-dependent oncoprotein EBNA-3C from EBV utilizes phosphorylation and disorder in pRb interaction. Known domains for the oncogenic protein EBNA-3C from EBV overlaid on results from PONDR analysis demonstrating regions of potential disorder (PONDR score greater than 0.5) with 66.8% predicted disorder (light blue). (**A**) EBNA-3C from EBV binds pRb through a potential LxCxE-mimic sequence at the N-terminus (purple) near the serine at position 134 (red), which is predicted to be phosphorylated by CK2, highlighted by a box (**B**) Zoomed in region within the box highlighted in Figure 7A.

**Figure 8 viruses-17-00835-f008:**
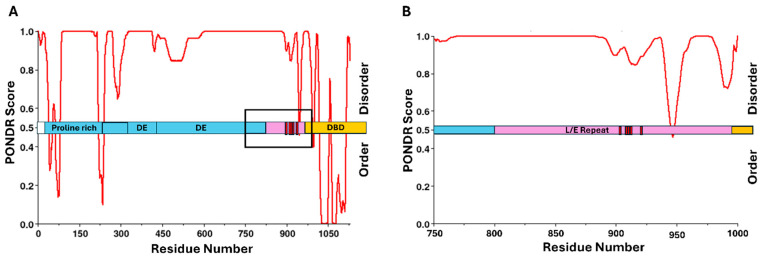
The oncoprotein LANA from KSHV utilized disorder and interacts with Rb independent of an LxCxE. The regions shown to interact with pRb are highlighted in purple, with regions of known or predicted disorder highlighted in light blue, and labels corresponding to known domains. (**A**) The pRb-binding domain for LANA from KSHV was shown to include residues 809–990 with no LxCxE/mimic domain thus far having been identified [191,240,248,249]. PONDR analysis predicted the highest level of disorder for all the proteins in this study, at 86.7%. Putative phosphoacceptor sites threonine 899 and serines 909, 910, 911, and 923 are depicted as red lines and the region is highlighted by a box. (**B**) Zoomed in region within the box highlighted in Figure 8A.

**Figure 9 viruses-17-00835-f009:**
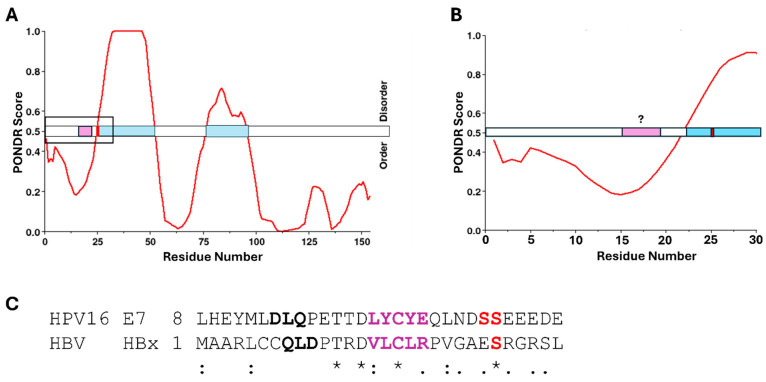
The oncoprotein HBx from HBV utilizes disorder and phosphorylation. The regions hypothesized to interact with pRb are highlighted in purple, with regions of known or predicted disorder highlighted in light blue. (**A**) HBx is not known to bind pRb directly, but has a potential LxCxE-mimic region in the N-terminus, shown in purple. PONDR analysis predicted 30.5% disorder and the phosphoacceptor serine residue is shown in red and highlighted by a box [263]. (**B**) Zoomed into the region within the box highlighted in Figure 9A with the potential LxCxE-mimic domain labeled with a question mark since it is a hypothesized interaction at this stage. (**C**) Clustal Omega multiple sequence alignment (https://www.ebi.ac.uk/jdispatcher/msa/clustalo, accessed on 23 April 2025) results comparing HBx to the disordered domain of E7 from high-risk HPV16 highlighting fully conserved (*), strongly similar (:) and weakly similar (.) amino acid residues between the sequences. The known phosphoacceptor serine residues from E7 are shown in red, along with the corresponding potential CK1 phosphoacceptor serine of HBx. An additional disordered region of similarity is highlighted in bold.

**Table 1 viruses-17-00835-t001:** Viruses defined as Group 1, carcinogenic to humans, or Group 2A/B, probably carcinogens, on the IARC list, including their respective oncoproteins known to dysregulate pRb, specific mode of pRb interaction, potentailly related sites of phosphorylation and the kinase responsible for phosphorylating those particular sites.

Virus	Abbreviation	Virus Family	Oncoprotein Causing pRb Dysregulation	Mode of pRb Interaction	Phorphorylation Site(s) Proximal to pRb Binding	Kinase
Human Papillomavirus	HPV	Papillomaviridae	E7	LxCxE	S31/S32	CK2
Merkel Cell Polyomavirus	MCPyV	Polyomaviridae	LTAg	LxCxE	S220	Unknown
BK Polyomavirus	BKPyV	Polyomaviridae	LTAg	LxCxE	S114	CK2 *
Human T-cell Lymphotropic Virus-1	HTLV-1	Retroviridae	Tax	LxCxE-mimic	S301/S302	Unknown
Hepatitis C Virus	HCV	Hepaviridae	NS5B	LxCxE-mimic	S326	Akt kinase
Epstein-Barr Virus	EBV (HHV-4)	Herpesviridae	EBNA-3C	LxCxE-mimic	S134 *	CK2 *
Kaposi Sarcoma-associated Herpesvirus	KSHV (HHV-8)	Herpesviridae	LANA	LxCxE-Independent	Unknown	NA
Hepatitis B Virus	HBV	Hepaviridae	HBx ?	Unknown	S25 ?	Unknown

* Putative, based on prediction data. ? Suspected, but has not been experimentally determined to date.

**Table 2 viruses-17-00835-t002:** Each oncogenic virus is listed with its abreviation, family, and oncogenic protein. The NCBI accession number corresponding to the amino acid sequence used both PONDR and NetPhos3.1 analysis within this review has also been included, along with the overall percentage of predicted disorder for each protein calcuated by PONDR analysis.

Virus	Abbreviation	Virus Family	Viral Oncoprotein	NCBI Accession	% Predicted Disorder
Human Papillomavirus	HPV	Papillomaviridae	E7 (HPV16)	NP_041326.1	27.6
Merkel Cell Polyomavirus	MCPyV	Polyomaviridae	LTAg	YP_009111421.1	32.7
BK Polyomavirus	BKPyV	Polyomaviridae	LTAg	PY_717940.1	24.5
Human T-cell Lyphotropic Virus-1	HTLV-1	Retrovridae	Tax	NP_957864.1	13
Hepatisis C Virus	HCV	Hepaviridae	NS5B RNA dependent RNA polymerase	CAB46677.1 NS5B (2420-3010 of the polypeptide)	18.6
Epstein-Barr Virus	EBV (HHV-4)	Herpesviridae	EBNA-3C	YP_401671.1	66.8
Kaposi Sarcoma-associated Herpesvirus	KSHV (HHV-8)	Herpesviridae	LANA	YP_001129431.1	86.7
Hepatisis B Virus	HBV	Hepaviridae	HBx	ABG23459.1	30.5

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
