# Peer review of "The Use of Intrinsic Disorder and Phosphorylation by Oncogenic Viral Proteins to Dysregulate the Host Cell Cycle Through Interaction with pRb"

_viruses, 2025, doi:10.3390/v17060835_

Round 1
Reviewer 1 Report
Comments and Suggestions for Authors
Kast-Woelbern et al., in their review entitled “The use of intrinsic disorder and phosphorylation by oncogenic viral proteins to dysregulate the host cell cycle through interaction with pRb,” provide a comprehensive overview of human oncogenic viruses and the mechanisms by which they regulate the tumor suppressor pRb. More specifically, the authors focus on the viral proteins expressed by human tumor viruses that target pRb for inactivation, providing an in-depth discussion of their structural/pRb-binding motifs, intrinsic disorder, and phosphorylation. The authors introduce each virus with information including basic molecular virology, epidemiology, and their etiological role in human cancers. For each virus, the viral oncoproteins that target pRb are described as well as the binding motifs utilized for this association. The authors combine this analysis with PONDR (Predictor of Natural Disordered Regions) analysis to compare and contract these viral oncoproteins with respect to intrinsically disordered regions (IDRs). In doing so, they are able to highlight common strategies used by tumor viruses to disrupt host cell proliferation via pRb.
Overall, this is a very well-written review article that provides an excellent synopsis of human tumor viruses and their viral proteins that target pRb. It provides ample detail for each virus and nicely compares the specific viral proteins targeting pRb. It is informative and highlights a unique aspect of viral oncoprotein biology (IDR, phosphorylation) that has rarely been focused on in other reviews. It will be an excellent review for anyone performing comparative analyses of human tumor virus action during carcinogenesis.
I only have a few minor corrections/suggestions:
- Line 38 and throughout: The correct acronym is “MCPyV”, not “McPyV.” The authors adopt the correct acronym later in the manuscript, but its usage should be consistent (including in Table 1).
- Line 205: ‘transient’ is misspelled.
- Lines 260-261: Hesbacher et al. have reported that MCPyV LT binds specifically to pRb, and not to p107 and p130. Also, ‘p170’ should be corrected to ‘p107.’
- Figure 2: It might be useful to include and/or discuss PONDR analysis of the truncated LT found in MCC tumors. Alternatively, it would be useful to indicate in the figure where the truncations occur so readers can visualize where within the protein this occurs w.r.t. intrinsic disorder.
- Table 2 currently satisfies the need for a summary figure, but an ‘overview’ visual that consolidates the information presented (all viral proteins in one figure, themes, binding motifs to target Rb, etc) would also be helpful.
Author Response
Please see the attached pdf for response to Reviewer 1. Thank you.

Reviewer 2 Report
Comments and Suggestions for Authors
Kast-Woelbern et al have attempted to provide a broad review of the interactions of viral proteins from human tumor viruses on pRb function.
While this is a pretty good idea for a review, the depth of the knowledge comes across as rather shallow despite the rather long nature of the review. The focus on intrinsic disorder and phosphorylation comes across as somewhat contrived. Additionally, a number of statements are too simplistic as stated and or just wrong as written. Some real details of existing pRB/LxCxE interactions with a strong structural focus would make this a much more impactful review.
Major Points:
1) the manuscript needs much more detail of a structural basis of LxCxE interaction with pRb based on say HPV E7 interaction, which could then be extended to other viral proteins with LxCxE motifs.
2) It should be clearly stated and illustrated with a figure that viral proteins have found multiple ways to inactive pRb. These include: 1) direct competition to free E2F (really best illustrated by adenovirus E1A, which is not discussed), 2) degradation of pRb so its just taken out of the regulatory system, or 3) directed inappropriate phosphorylation by a v-cyclin or overexpressed/activate cellular CDK. There may be other mechanisms that I missed that could be incorporated.
3) The emphasis on intrinsic disorder is highly contrived as presented. I would recommend removing those figures which are simply based on predictive software like PONDR. Why is this so important if a bunch of them are actually predicted to be ordered vs disordered? Disorder increases the ability sample conformational space and the potential number of factors that can be bound but reduces the overall strength of the interaction as the entropic cost of the interaction subtracts from the enthalpic gains of the new bonds formed by the protein interaction.
4) The phosphorylation focus also comes across as contrived as written. What are the likelihoods of finding a potential phosphoserine in the vicinity of any motif? Probably pretty high. How is this above random chance? Again most of this type of analysis comes across as predictive "hand waving", not based on actual data or in line with the description of an experimentally defined structural role that is well understood.
Minor points:
1) Most of the HPV E7 data is related to HPV16, which is indeed 98 amino acids in length. HPV18 E7 is actually 101 aa. The authors need to be careful when writing about HPV E7s in general and E7 from a specific HPV type to make this clear every time. Otherwise statements like line 177 are just wrong.
2) I am not aware of any evidence that pRb is in the cytoplasm and blocks E2F entry into the nucleus (line 85 and 91). This contradicts the next sentence that says pRb recruits HDACs to promoter bound E2F to repress S-phase specific gene expression.
3) As written on line 172, it sounds like cell cycle is halted by the process of infection with HPV and that viral products like E6 and E7 release this infection induced arrest. This has no relationship to the reality that E6 and E7 force naturally quiescent cells to divide inappropriately.
4) High risk HPV E7 target pRb for degradation. This was well established by K. Munger's lab and needs to be mentioned.
5) Acid amino acids adjacent to the LxCxE in HPV16 E7 likely contribute to pRB interaction based on mutational studied done by Dick et al (PMID: 12021356). This extends the interaction beyond the structure solved by Pavletich (PMID: 9495340). There is other real structural information on this interaction that should be carefully curated and used to put the interactions with other viral proteins in context.
6) I am not aware of any human polyomaviruses that integrate into the human genome as a normal part of their replicative cycle ( line 226).
7) Line 233 - tissue samples are not diagnosed with MCC. Patients are diagnosed with MCC. Rewrite to be correct, perhaps more like "MCPyV was first identified from tissue samples from patients diagnosed with MCC.
8) Lines 266-267 is not clear as written
9) What about HTLV-2? Its oncogenic too.
10) Line 501 - KSHV is not defined until later
11) line 71 - mention that integration leads to disregulated higher expression of E6 and E7 generally due to the loss of repression by E2 that alter cell cycle changes more aggressively than found in the normal infection.
12) line 154 - not sure all HPVs have 8 protein coding genes.
13) the numbers on lines 10 and 32 seem inconsistent
Author Response
Please see the attached pdf for response to Reviewer 2. Thank you.

Round 2
Reviewer 2 Report
Comments and Suggestions for Authors
The authors have taken my comments and suggestion seriously and appear to have worked diligently to address them. They also repaired a number of other important issues that I originally missed, presumably based on comments from other reviewers. This review has a wider scope than most and touches on a lot of material, which is always a challenge, and I think they have done a reasonable job organizing and presenting the material.